# The Role of Nutrient Supplements in Female Infertility: An Umbrella Review and Hierarchical Evidence Synthesis

**DOI:** 10.3390/nu17010057

**Published:** 2024-12-27

**Authors:** Chhiti Pandey, Alison Maunder, Jing Liu, Vaishnavi Vaddiparthi, Michael F. Costello, Mahnaz Bahri-Khomami, Aya Mousa, Carolyn Ee

**Affiliations:** 1NICM Health Research Institute, Western Sydney University, Penrith, NSW 2751, Australia; gp4211765@gmail.com (C.P.); alison.maunder@westernsydney.edu.au (A.M.); jing.liu@westernsydney.edu.au (J.L.); vaishv0810@gmail.com (V.V.); 2Women’s Health, University of New South Wales and Royal Hospital for Women and Monash IVF, Sydney, NSW 2031, Australia; mfcostello@unsw.edu.au; 3Monash Centre for Health Research and Implementation, Monash University, Melbourne, VIC 3168, Australia; mahnaz.bahrikhomami@monash.edu (M.B.-K.); aya.mousa@monash.edu (A.M.)

**Keywords:** female infertility, female fertility agents, assisted reproductive techniques, dietary supplements, pregnancy rate, pregnancy complications, umbrella review

## Abstract

Background and Objectives: Nutrient supplements are commonly used to improve fertility outcomes by women with infertility trying to conceive spontaneously or utilising medically assisted reproduction (MAR). However, despite their widespread use and perceived safety, there is a lack of clear guidance on the efficacy and safety of these supplements for female infertility. The aim of this umbrella review was to identify the best available and most recent evidence on the efficacy and safety of nutrient supplements for female infertility to provide evidence-based guidance for clinicians and reproductive couples. Methods: Five electronic databases were searched for umbrella reviews, meta-analyses, and systematic reviews of randomised controlled trials on nutrient supplements for female infertility, published from August 2017 to January 2024. The primary outcomes were live birth, and clinical and biochemical pregnancy rates. Secondary outcomes were adverse effects including miscarriage and ectopic or multiple pregnancy. Quality assessment was performed using the A MeaSurement Tool to Assess systematic Reviews Version 2.0 (AMSTAR 2), and the certainty of evidence for outcomes were assessed using the Grading of Recommendations Assessment, Development and Evaluation (GRADE) approach, where possible. Results: Four meta-analyses were included. Multiple micronutrients and antioxidants increased live birth rates in women utilising MAR and/or trying to conceive spontaneously compared to placebo, standard or no treatment (odds ratio (OR) 2.59 and 1.81 respectively) with very low certainty evidence. L-carnitine, coQ10, melatonin, myo-inositol, NAC and vitamin D increased clinical pregnancy rates in women with PCOS and/or undergoing MAR compared to placebo, standard or no treatment (odds ratio (OR) 11.14, 2.49, 1.66, relative risk (RR) 1.52, OR 2.15, and 1.49 respectively) with very low certainty evidence. Vitamin D did not increase biochemical pregnancy rates in women utilising MAR with very low certainty evidence. NAC, vitamin D, and pooled antioxidants had no effect on miscarriage rates or multiple pregnancy rates in women trying to conceive spontaneously or utilising MAR, with low to very low certainty evidence. Pooled antioxidants had no effect on ectopic pregnancy rates in women trying to conceive spontaneously or utilising MAR, with low certainty evidence. Conclusions: The available evidence is insufficient to recommend nutrient supplementation to improve female infertility in women trying to conceive naturally and those utilising MAR. However, there is currently no indication that these nutrients pose any risk of significant harm. Registration: PROSPERO (CRD42022365966) 20 October 2022.

## 1. Introduction

Infertility is a disease of the female or male reproductive system characterised by the inability to conceive after one year or more of regular unprotected sexual intercourse [1] as defined by the World Health Organization (WHO). Infertility affects an estimated 17.5%, or one in six individuals, globally [1]. Female infertility is estimated to contribute to one-third of cases, while male fertility accounts for another third, and the remaining cases attributed to both sexes or unknown factors [2]. Infertility is classified as primary (never having achieved a clinical pregnancy) or secondary infertility (having achieved at least one prior clinical pregnancy) [3]. The impact of infertility has significant societal and health consequences including mental health impacts [4], social stigma [5], economic hardship [6], and gender-based violence [7].

The most common causes of female infertility are ovulatory dysfunction and tubal disease, accounting for 25% and 11–67% of infertility diagnoses, respectively [8,9]. Polycystic ovary syndrome (PCOS) is the most common cause of anovulatory infertility, affecting 80% of cases [10]. Tubal infertility results from blocked fallopian tubes or pelvic adhesions, commonly caused by sexually transmitted infections [9]. Endometriosis, the presence of endometrial-like tissue outside the uterine cavity, can also cause pelvic adhesions through intra-abdominal inflammation and scar tissue [11]. Endometriosis affects 7–10% of females of reproductive age [12] and up to 40% of those experiencing infertility [13]. Other gynaecological conditions contributing to female infertility include premature ovarian insufficiency [14], pelvic inflammatory disease [15], and uterine and cervical factors, such as endometrial polyps, uterine fibroids, and post-surgery scarring [2,16]. Additionally, female infertility increases with age, from a prevalence of 1% at age 25 to 55% at age 45 [17]. Lifestyle factors negatively impacting fertility include poor diet [18], obesity, stress, smoking, and alcohol consumption [2].

Treatment of female infertility varies based on the diagnosis and may include expectant management, surgery, or medically assisted reproduction (MAR) [19]. Surgery is invasive, while expectant management may be ineffective where medical intervention is indicated. MAR treatments, which may include ovulation induction, assisted insemination and in vitro fertilisation (IVF), are expensive [20], with inequity of access [21], and an estimated 29% of treatments result in live births following a fresh transfer cycle [22]. Additionally, there are risks, such as multiple pregnancies, which occur in up to 36% in women taking gonadotropins [23]. Together, these challenges highlight the need to identify complementary therapies which may serve as low-cost adjuncts for improving female fertility.

Many individuals with female infertility turn to complementary therapies, including nutrient supplements, when trying to conceive spontaneously or undergoing MAR [24]. Nutrient supplementation during preconception (defined as the three months before conception) is increasingly common with between 50% and 70% of women in Australia, Singapore and the Netherlands taking supplements during this time [25,26,27]. Similarly, a UK survey found that over 70% of women starting MAR were taking nutrient supplements [28]. There is growing evidence that nutrient supplements may positively influence pregnancy and live birth rates [29,30]. Hence, the global supplement market, valued at over US$382 billion, is estimated to nearly double by 2030 [31], with women being the dominant consumers [32].

Despite widespread use and a general perception of safety of nutrient supplements among individuals of reproductive age, there is a lack of clear guidance for clinicians and reproductive-aged couples regarding the efficacy and safety of nutrient supplements for female infertility [33]. National guidelines from the Netherlands, USA and Italy, and also WHO guidelines [34], have made recommendations for improving preconception health, however, supplementation is limited to folic acid [35], intended to prevent neural tube disorders rather than to assist conception [36]. Similarly, previous systematic reviews have primarily focused on single nutrients or categories of nutrients, with no umbrella reviews to summarise the clinical efficacy of nutrient supplements overall.

In light of this shortcoming, we undertook an umbrella review with the aim of identifying the best available and most recent evidence from systematic reviews, meta-analyses, network meta-analyses or umbrella reviews on the efficacy and safety of nutrient supplements to improve fertility outcomes in individuals with female infertility trying to conceive spontaneously or utilising MAR.

## 2. Materials and Methods

This umbrella review (also known as an overview of reviews) was conducted in accordance with the Preferred Reporting Items for Overviews of Reviews (PRIOR) guidelines [37]. The review protocol was developed a priori and prospectively registered on PROSPERO (CRD42022365966) on 20 October 2022. No changes were made to the protocol after registration.

### 2.1. Search Strategy

A preliminary scoping search was conducted to identify reviews on the topic, noting text words and index terms in relevant reviews. A comprehensive search strategy was then developed, limited to English language articles and a five-year timespan to identify the most recent evidence. The full search strategy is available in Appendix A. Databases included AMED, CINAHL, All EBM Reviews, MEDLINE and Embase, which were initially searched from August 2017 to August 2022 and then updated in January 2024 to capture more recent reviews. Titles and abstracts were screened by two of the four authors (C.P., J.L., V.V. and A.M. (Alison Maunder)) independently using our selection criteria below. Two of the four authors (C.P., J.L., V.V. and A.M. (Alison Maunder)) then independently assessed full-text articles. Any disagreements were resolved by either consensus or discussion with a third author (C.E.).

### 2.2. Selection Criteria

Study Design: All English-language umbrella reviews, meta-analyses (MA), and systematic reviews (SR) of data from randomised controlled trials (RCTs) were considered. Reviews were only included if they were published in the last five years from the search date, had clear inclusion criteria, reported a systematic search and data extraction procedure, and conducted a quality assessment. Reviews were excluded if they included experimental or animal studies.

Selection criteria were applied according to the Population, Intervention, Control and Outcome (PICO) framework developed a priori in our protocol and outlined below.

Population: females aged 18–45 years with primary or secondary infertility of any duration and cause were included. Individuals trying to conceive spontaneously or utilising MAR were included. Reviews that included studies evaluating couple infertility and male infertility were excluded unless a subgroup analysis was available for female infertility only.

Intervention: Nutrient supplements were defined as single nutrients or combinations of vitamins, minerals, pre/probiotics, fatty acids, or amino acids (including their precursors) provided in supplement form and administered orally. Foods were excluded, with the exception of the addition of a nutrient supplement to a food (e.g., yoghurt with added probiotics). Reviews of dietary modification interventions and herbal supplements were excluded.

Comparison: Any comparative control including placebo, no treatment, or other nutrient supplements or interventions (as long as there was a direct comparison of the intervention to a control) were included. Control groups where the effects of the intervention could not be isolated were excluded.

Outcomes: Primary outcome measures were live births, clinical pregnancy (confirmed by gestational sac formation or foetal heart activity on ultrasound) or biochemical pregnancy (confirmed by a positive result from a urine test, blood test or both). Any adverse effects reported by the review including miscarriage and ectopic or multiple pregnancy. Any review whose outcomes were not listed above were excluded.

### 2.3. Data Collection and Analysis

#### 2.3.1. Hierarchical Evidence Gathering

We followed a systematic approach towards data collection to prioritise the top tiers of evidence, as previously published by our team [38,39]. The hierarchy of evidence considered in the selection process was structured as follows: (1) umbrella reviews; (2) network meta-analyses of RCTs; (3) pair-wise meta-analyses of double-blind placebo controlled RCTs; (4) pair-wise meta-analyses of RCTs; (5) systematic reviews of RCTs (without meta-analyses). Where multiple reviews overlapped, the review with the most recent search end date was included. Where both an umbrella review and a systematic review and meta-analysis addressed the same PICO criteria, the umbrella review was prioritised for inclusion. If the systematic review and meta-analysis was more recent, umbrella review findings were updated with the more recent data. In situations where the degree of overlap was uncertain, a citation matrix was used to calculate the corrected covered area and provide an estimate of the overlap between primary studies [40]. If the overlap was considered high (>15%), the most recent, and then the most comprehensive review was selected.

#### 2.3.2. Data Extraction

Two reviewers (C.E. and C.P.) independently extracted data into an extraction template developed specifically for the study using Microsoft Excel. The data extracted included: review type, dates of literature searches, the number of trials and participants, sample characteristics and intervention and control details. Outcomes were extracted as weighted or standardised mean differences and confidence intervals for continuous outcomes, odds ratios or risk ratios for dichotomous outcomes. Quality assessments including risk of bias and Grading of Recommendations Assessment, Development and Evaluations (GRADE) were extracted when available.

### 2.4. Quality Assessment

Quality assessments of the included reviews were completed using the AMSTAR (A Measurement Tool to Assess Systematic Reviews) Version 2.0 tool [41]. AMSTAR 2 comprises 16 items for critical appraisal of systematic reviews, with critical domains described in Appendix A. Two reviewers (C.P. and M.B.-K. or A.M. (Alison Maunder)) independently applied the checklist for each study review, and any disagreement was resolved with discussion. Using this tool, each review received an overall confidence rating based on four categories: “high” representing no critical flaws and ≤1 non-critical flaw; “moderate” representing no critical flaws but more >1 non-critical flaw; “low” representing ≤1 critical flaw with or without non-critical flaws; and “critically low” representing >1 critical flaw with or without non-critical flaws.

### 2.5. Certainty of Evidence

For the assessment and reporting of the certainty of evidence (confidence in effect-estimates), the GRADE tool [42] was used for the outcomes of live birth, clinical and biochemical pregnancy, and adverse effects. Where study authors conducted a GRADE assessment for these outcomes, that assessment was extracted directly. Otherwise, GRADE assessments were conducted independently by two of the four authors (C.P., A.M. (Aya Mousa), C.E. and M.B.-K.). Discrepancies, if any, were resolved by discussion. Not all reviews reported enough information to conduct a GRADE assessment or only included one RCT in the outcome analysis.

## 3. Results

### 3.1. Search Results

The search yielded 3842 results, with 2378 remaining after the removal of duplicates. Following title and abstract screening, 128 studies remained. After full text screening, a total of 16 studies were eligible to be included. Of these, 12 studies were excluded after the hierarchical evidence synthesis screening, and four reviews were eligible to be included in our review (Figure 1). The excluded studies and the reason for exclusion are listed in Appendix A.

### 3.2. Study Characteristics

All four reviews included were pair-wise meta-analyses conducted between 2019 and 2022 and published between 2020 and 2023. Reviews ranged in size from nine studies to 63 RCTs. The characteristics and outcomes of included reviews are reported in Table 1.

#### 3.2.1. Population

All reviews included women of reproductive age, defined as those between the ages of 18 and 45. Two reviews included those with subfertility attending fertility clinics that might or might not undergo MAR [43,44], one included those with infertility undergoing MAR [45], and one included those with PCOS undergoing MAR [46].

#### 3.2.2. Interventions

The included reviews examined inositol [46], *N*-acetylcysteine (NAC) [43], vitamin D [45], or a range of oral antioxidant supplements which included NAC and vitamin D as well as L-arginine, L-carnitine, coenzymeQ10 (coQ10), melatonin, vitamin B complex, vitamin C, vitamin E, and multiple micronutrients [44].

#### 3.2.3. Control

All interventions were compared with placebo and/or no treatment. A comparison of lower and higher doses of melatonin was reported in one review [44]. In addition, two reviews compared nutrients to metformin [43,46]. Folic acid at doses less than 1 mg were considered ‘standard treatment’ in Showell et al., 2020 [44].

#### 3.2.4. Outcomes

All four reviews reported on clinical pregnancy rates [43,44,45,46], one reported on live birth rates [44] and one reported on biochemical pregnancy rates [45]. Multiple pregnancy and miscarriage rates were reported by three reviews [43,44,45], while ectopic pregnancy was reported by one review [44]. One review reported these adverse events for pooled interventions but not by individual nutrient [44].

### 3.3. Quality Assessment Using AMSTAR 2

One review [44] demonstrated no critical weaknesses, whereas the remaining three reviews [43,45,46] each had three or more critical weaknesses. The most common critical weaknesses included failing to provide a list of justifications for excluded studies (item 7) and consider risk of bias when interpreting the results of the review (item 13). Table 2 provides a summary of the AMSTAR 2 assessments.

**Figure 1 nutrients-17-00057-f001:**
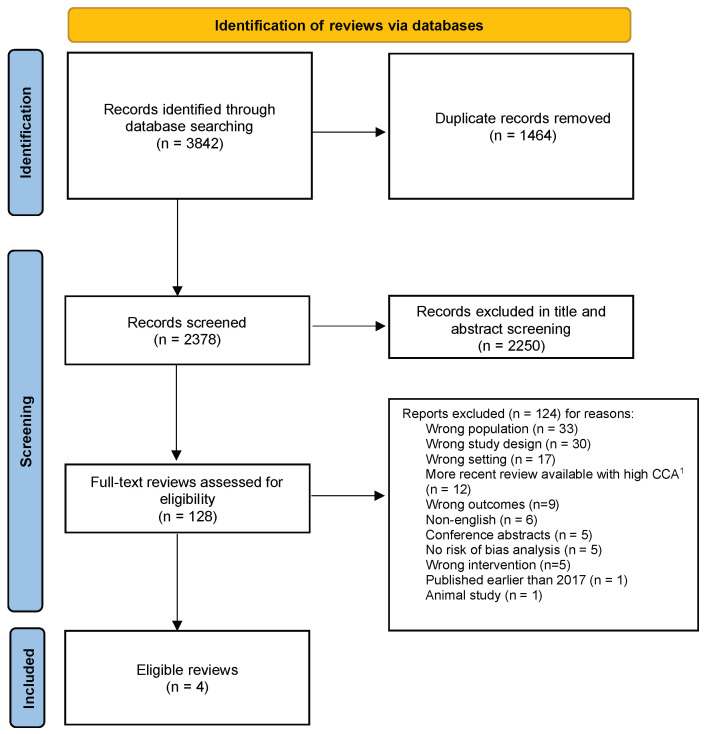
PRISMA diagram of review selection process [47]. ^1^ Corrected covered area: degree of overlap between reviews.

**Table 1 nutrients-17-00057-t001:** Characteristics and main findings of the included reviews of nutrients for female infertility.

Author and Year (Review Type): Search Date	No. of Participants (Trials) Inclusion Criteria	Intervention (*Control*)	Outcomes—Live Birth (LBR) (GRADE Assessment by: + Original Author, * Umbrella Review Team, ^ No GRADE)	Outcomes-Clinical Pregnancy/Biochemical Pregnancy (GRADE Assessment by: + Original Author, * Umbrella Review Team, ^ No GRADE)	Outcomes—Adverse Events
**AMINO ACIDS**					
**L-ARGININE**					
Showell et al., 2020 [44] (SRMA) Inception—April 2013 and updated in September 2019.	N = 7760 (63 RCTs; 7 RCTs, N = 678 for L-arginine) P = women undergoing MAR I = L-arginine C = placebo, no treatment O = live birth, clinical pregnancy	L-arginine 16 g/day for 10–12 days/cycle (*Placebo or no treatment*)	Live birth NS (1 RCT, N = 37) ^	Clinical pregnancy NS (2 RCTs, N = 71) ⨁◯◯◯* VERY LOW certainty (very serious inconsistency, very serious imprecision)	NA
**L-CARNITINE**					
Showell et al., 2020 [44] (SRMA) Inception—April 2013 and updated in September 2019.	N = 7760 (63 RCTs; 2 RCTs, N = 450 for L-carnitine) P = women at fertility clinic with clomiphene resistant PCOS I = L-carnitine C = placebo O = clinical pregnancy	Carnitine 3 g/day from day 3 until the day of first positive pregnancy test (*Placebo*)	NR	↑Clinical pregnancy (OR 11.14, 95% CI 5.70 to 21.81; N = 450: 2 RCTs) ⨁◯◯◯* VERY LOW certainty (very serious inconsistency, very serious imprecision)	NA
**ANTIOXIDANTS**					
**CO-ENZYME Q10 (CoQ10)**					
Showell et al., 2020 [44] (SRMA) Inception—April 2013 and updated in September 2019.	N = 7760 (63 RCTs; 4 RCTs, N = 397 for CoQ10) P = women at fertility clinic/clomiphene resistant PCOS I = CoQ10 C = placebo, no treatment O = live birth, clinical pregnancy	CoQ10 Dose: 180–600 mg/day (*Placebo or no treatment*)	Live birth NS (2 RCTs, N = 225) ⨁◯◯◯* VERY LOW certainty (very serious ROB, very serious imprecision)	↑Clinical pregnancy (OR 2.49, 95% CI 1.50 to 4.13, N = 397: 4 RCTs) ⨁◯◯◯* VERY LOW certainty (very serious ROB, serious inconsistency, serious imprecision)	NA
**MELATONIN**					
Showell et al., 2020 [44] (SRMA) Inception—April 2013 and updated in September 2019.	N = 7760 (63 RCTs; 7 RCTs, N = 678 for melatonin) P = women at fertility clinic with various reasons for infertility I = melatonin C = placebo, no treatment, higher dose melatonin O = live birth, clinical pregnancy	Melatonin 3–16 mg/day (*Placebo, no treatment, melatonin*)	**Subgroup analyses by control** *Melatonin* vs. *placebo/no treatment* Live birth NS (2 RCTs, N = 270) ⨁◯◯◯* VERY LOW certainty (serious ROB, very serious imprecision) *Melatonin lower dose* vs. *melatonin higher dose* Live birth NS (2 RCTs, N = 140) ⨁◯◯◯* VERY LOW certainty (serious ROB, very serious imprecision)	**Subgroup analyses by control** *Melatonin* vs. *placebo/no treatment* ↑Clinical pregnancy (OR 1.66, 95% CI 1.12 to 2.47, N = 678: 7 RCTs) ⨁◯◯◯* VERY LOW certainty (very serious ROB, serious inconsistency, serious imprecision) *Melatonin lower dose* vs. *melatonin higher dose* Clinical pregnancy NS (2 RCTs, N = 140) ⨁◯◯◯* VERY LOW certainty (serious ROB, very serious imprecision)	NA
**MYO-INOSITOL**					
Unanyan et al., 2022 [46] (SRMA) Inception—November 2021	N = 4668 (35 RCTs) P = women with PCOS undergoing MAR including OI/IVF/IUI/ICSI I = metformin and/or myo-inositol C = folic acid, myo-inositol, no treatment, placebo O = clinical pregnancy, ovarian hyperstimulation, live birth	Myo-inositol 4 g/day (*Metformin, placebo or no treatment*)	NR	**Subgroup analyses by control** *Myo-inositol* vs. *metformin* ↑Clinical pregnancy (RR = 1.52, 95% CI: 1.05 to 2.18, N = 220: 2 RCTs) ⨁◯◯◯* VERY LOW certainty (very serious inconsistency, very serious imprecision) *Myo-inositol* vs. *placebo or no treatment* Clinical pregnancy NS (2 RCTs, N = 236) ⨁⨁◯◯* LOW certainty (serious inconsistency, serious imprecision)	NR
** *N* ** **-ACETYLCYSTEINE (NAC)**					
Showell et al., 2020 [44] (SRMA) Inception—April 2013 and updated in September 2019.	N = 7760 (63 RCTs; 1 RCT, N = 60 for NAC) P = women with PCOS undergoing ovarian drilling I = NAC C = placebo O = live birth	NAC 1200 mg/day for 5 days from day 3 for 12 cycles (*Placebo*)	↑Live birth (OR 3.00, 95% CI 1.05 to 8.60; N = 60: 1 RCT) ^		NA
Devi et al., 2021 [43](SRMA) Inception—April 2019	N = 2233 (13 RCTs) P = women visiting fertility clinics, might/might not undergo MAR aged 18–45 I= NAC C = another adjuvant therapy/placebo/no treatment O = clinical pregnancy, adverse events	NAC 1200 mg/day: -2 doses 600 mg-8 RCTs -3 doses 400 mg-1 RCT NAC 1800 mg/day: -3 doses 600 mg-6 RCTs (*Metformin/placebo/no treatment*)	NR	**Pooled analysis** Clinical pregnancy NS (13 RCTs, N = 2233) ⨁◯◯◯* VERY LOW certainty (very serious ROB, very serious inconsistency) **Subgroup analyses by control** *NAC* vs. *Metformin* Clinical pregnancy NS (5 RCTs, N = 510) ⨁◯◯◯* VERY LOW certainty (serious ROB, very serious inconsistency, serious imprecision) *NAC* vs. *no treatment* ↑Clinical pregnancy (OR 2.15, 95% CI 1.01 to 4.60, N = 177: 2 RCTs) ⨁⨁⨁◯* MODERATE certainty (serious imprecision) *NAC* vs. *placebo* ↑Clinical pregnancy (OR 2.14, 95% CI 1.05 to 4.37, N = 1601: 8 RCTs) ⨁◯◯◯* VERY LOW certainty (serious ROB, serious indirectness, very serious imprecision)	**Pooled analysis** Multiple pregnancy NS (6 RCTs, N = 1474) ⨁◯◯◯* VERY LOW certainty (very serious ROB, serious indirectness, serious imprecision) Miscarriage rate NS (6 RCTs, N = 1367) ⨁◯◯◯* VERY LOW certainty (serious ROB, serious indirectness, serious imprecision) **Subgroup analyses by control** *NAC* vs. *Metformin* Multiple pregnancy NS (1 RCT, N = 192) ^ Miscarriage rate NS (1 RCT, N = 192) ^ *NAC* vs. *no treatment* ↑Multiple pregnancy (OR 3.00, 95% CI 0.12 to 75.48, N = 97: 1 RCT) ^ Miscarriage rate NS (1 RCT, N = 97) ^ *NAC* vs. *placebo* Multiple pregnancy NS (4 RCTs, N = 1474) ⨁◯◯◯* VERY LOW certainty (serious ROB, serious indirectness, very serious imprecision) Miscarriage rate NS (4 RCTs, N = 1078) ⨁◯◯◯* VERY LOW certainty (serious ROB, serious indirectness, serious imprecision)
**VITAMINS**					
**VITAMIN B COMPLEX**					
Showell et al., 2020 [44] (SRMA) Inception—April 2013 and updated in September 2019.	N = 7760 (63 RCTs; 1 RCT, N = 102 for B-complex vitamins) P = women with infertility and insulin-resistant PCOS undergoing OI, IVF or ICSI I = vitamin B complex C = no treatment O = live birth, clinical pregnancy	I: Vitamin B complex (50 mg B6, 400 mcg folic acid, 500 mcg B12, 1 g trimethylglycine, 6 mg pyridoxal-5-phosphate): 1 tablet/day for 3 cycles (*400 mcg folic acid*)	Live birth rate NS (1 RCT, N = 102) ^	Clinical pregnancy NS (1 RCT, N = 102) ^	NA
**VITAMIN C**					
Showell et al., 2020 [44] (SRMA) Inception—April 2013 and updated in September 2019.	N = 7760 (63 RCTs; 2 RCTs, N = 899 for vitamin C) P = women with subfertility or endometriosis at fertility clinic I = vitamin C C = placebo/no treatment O = clinical pregnancy	I: Vitamin C/ascorbic acid 1–10 g/day (*Placebo, no treatment*)	NR	Clinical pregnancy NS (2 RCTs, N = 899) ⨁◯◯◯* VERY LOW certainty (very serious ROB, serious inconsistency, very serious imprecision)	NA
**VITAMIN D**					
Showell et al., 2020 [44] (SRMA) Inception—April 2013 and updated in September 2019.	N = 7760 (63 RCTs; 2 RCTs, N = 92 for vitamin D) P = women with subfertility attending fertility clinic, might/might not undergo MAR I = vitamin D C = placebo O = live birth, clinical pregnancy, adverse events	Vitamin D (*Placebo*)	Live birth rate NS (1 RCT, N = 52) ^		NA
Meng et al., 2023 [45] (SRMA) Inception—March 2022.	N = 2352 (9 RCTs) P = women with infertility undergoing IVF, ICSI, ET I = vitamin D C = placebo, sunflower seed oil, standard or no treatment O = biochemical and clinical pregnancy, miscarriage rate, multiple pregnancy.	Vitamin D 2000–600,000 IU (*Placebo, sunflower seed oil, standard or no treatment*)	NR	Biochemical pregnancy NS (7 RCTs, N = 1483) ⨁◯◯◯* VERY LOW certainty (serious ROB, very serious imprecision) ↑Clinical pregnancy (OR 1.49, 95% CI 1.05 to 2.11, 9 RCTs, N = 1677) ⨁◯◯◯* VERY LOW certainty (serious ROB, very serious inconsistency, very serious imprecision)	Miscarriage rate NS (7 RCTs, N = 655) ⨁◯◯◯* VERY LOW certainty (serious indirectness, very serious imprecision) Multiple pregnancy NS (3 RCTs, N = 651) ⨁◯◯◯* VERY LOW certainty (very serious ROB, very serious inconsistency, very serious imprecision)
**VITAMIN E**					
Showell et al., 2020 [44] (SRMA) Inception—April 2013 and updated in September 2019.	N = 7760 (63 RCTs; 1 RCT, N = 107 for vitamin E) P = women with infertility undergoing OI/IUI I = vitamin E C = no treatment O = live birth, clinical pregnancy	I: Vitamin E 400 IU/day until hCG injection (*No treatment*)	Live birth rate NS (1 RCT, N = 107) ^	Clinical pregnancy NS (1 RCT, N = 107) ^	NA
**ANTIOXIDANT COMBINATIONS**					
Showell et al., 2020 [44] (SRMA) Inception—April 2013 and updated in September 2019.	N = 7760 (63 RCTs; 3 RCTs, N = 378 for combined antioxidants) P = women with subfertility undergoing MAR I = antioxidant combinations C = placebo with folic acid, no treatment O = live birth	Multiple micronutrients (MMN) 1 tablet/day for 3 menstrual cycles; Vitacap 1 capsule/day for 6 months; Seidivid 2 sachets/day for 2 months. (*Placebo or no treatment*)	↑Live birth (OR 2.59, 95% CI 1.52 to 4.40; N = 378: 3 RCTs) ⨁◯◯◯* VERY LOW certainty (serious ROB, very serious inconsistency, very serious imprecision)		NA
**POOLED ANTIOXIDANTS**					
Showell et al., 2020 [44] (SRMA) Inception—April 2013 and updated in September 2019.	N = 7760 (63 RCTs) P = women with subfertility attending fertility clinic, might/might not undergo MAR I = antioxidants, single or combinations (L-arginine, myo-inositol, carnitine, selenium, vitamin E, vitamin B complex, vitamin C, vitamin D + calcium, CoQ10, melatonin, folic acid, NAC, and omega-3-polyunsaturated fatty acids) C = placebo/no treatment/standard treatment O = live birth, clinical pregnancy, adverse events	I: Antioxidant(s) (L-arginine, myo-inositol, carnitine, selenium, vitamin E, vitamin B complex, Vitamin C, vitamin D + calcium, CoQ10, melatonin, folic acid, NAC, and omega-3-polyunsaturated fatty acids) (*Placebo/no treatment/standard treatment*)	**Pooled analysis** ↑Live birth (OR 1.81, 95% CI 1.36 to 2.43, N = 1227: 13 RCTs) ⨁◯◯+ VERY LOW certainty **Subgroup analyses by indication of subfertility**: *PCOS* ↑Live birth (OR 3.34, 95% CI 1.90 to 5.86, N = 362: 3 RCTs) *Tubal subfertility* NS (1 RCT, N = 37) *Unexplained subfertility* NS (2 RCTs, N = 133) *Poor ovarian reserve* NS (2 RCTs, N = 266) *Other indications for subfertility* ↑Live birth (OR 1.70, 95% CI 1.02 to 2.83, N = 338: 3 RCTs)	**Subgroup analyses by indication of subfertility** *PCOS * ↑Clinical pregnancy (OR 4.24, 95% CI 3.23 to 5.56, N = 1908: 16 RCTs) *Tubal subfertility* NS (2 RCTs, N = 71) *Unexplained subfertility* NS (4 RCTs, N = 997) *Poor responders* NS (1 RCT, N = 65) *Poor ovarian reserve* NS (2 RCTs, N = 266) *Endometriosis* NS (1 RCT, N = 280)	Miscarriage rate NS (24 RCTs, N = 3229) ⨁⨁◯◯+ LOW certainty Multiple pregnancy NS (9 RCTs, N = 1886) ⨁⨁◯◯+ LOW certainty Ectopic pregnancy NS (3 RCTs, N = 404) ⨁⨁◯◯+ LOW certainty

LEGEND: **CoQ10**, Coenzyme Q10; **ET**, embryo transfer; **ICSI**, Intracytoplasmic sperm injection; **IUI**, intrauterine insemination; **IVF**, In-vitro fertilisation; **MAR**, medically assisted reproduction; **mg**, milligram; **mcg,** microgram; **MMN**, multiple micronutrients containing thiamine, riboflavin, niacin B3, vitamins B6 and B12, folate, vitamins C, A and D, calcium, phosphorus, magnesium, sodium, potassium, chloride, iron, zinc, copper, selenium, iodine, vitamin E, vitamin K, L-arginine, inositol, NAC, biotin, pantothenic acid; **NA**, Not assessed; **NAC**, *N*-acetylcysteine; **NR**, Not reported; **NS**, Not significant; **OI**, ovulation induction; **PCOS**, Polycystic ovarian syndrome; **RCT**, randomised controlled trial; **ROB**, risk of bias; **Seidivid**, contains myo-inositol 2 g, 0.975 mg melatonin, 200 mcg folic acid, 27.5 mcg selenium; **Vitacap**, vitamin A (Palmitate) 5000 IU, vitamin B1 (thiamine mononitrate) 5 mg, vitamin B6 (pyridoxine HCL) 2 mg, vitamin B12 (cyanocobalamin 5 mg, vitamin C 75 mg, vitamin D3 (cholecalciferol) 400 IU, Vitamin E (d-alpha tocopheryl acetate) 15 mg, nicotinamide 45 mg, folic acid 1000 mcg, ferrous fumarate 50 mg, dibasic calcium phosphate 70 mg, copper sulphate 0.1 mg, manganese sulphate 0.01 mg, zinc sulphate 50 mg, potassium iodide 0.025 mg and magnesium oxide 0.5 mg. **GRADE** assessment by: + = original author, * = umbrella review team, ^ = no GRADE performed; ↑ = increase in outcome measure with intervention.

**Table 2 nutrients-17-00057-t002:** AMSTAR 2 rating of included reviews.

Nutrient Supplement	Reference	Critical Flaws ^a^	AMSTAR Rating ^b^
Various oral antioxidants	Showell et al., 2020 [44]	None	High
*N*-acetyl cysteine	Devi et al., 2021 [43]	7, 11, 13	Critically Low
Vitamin D	Meng et al., 2023 [45]	7, 13, 15	Critically Low
Inositol	Unanyan et al., 2022 [46]	7, 13, 15	Critically Low

^a^ Refers to the item numbers of the AMSTAR 2 critical domains list which the review has failed to satisfy. ^b^ Refers to the AMSTAR 2 overall confidence rating in the results of the review: High, Moderate, Low, Critically low.

### 3.4. Efficacy of Nutrient Supplementation for Live Birth Rate

L-arginine, coQ10, melatonin, NAC, vitamin B complex, vitamin D, vitamin E and multiple micronutrients were assessed for increased live birth rate in RCTs when compared with placebo or no treatment. Only three comparisons (multiple micronutrients, coQ10, melatonin) combined more than one trial. Supplements containing multiple micronutrients increased live birth rate in women undergoing MAR compared with placebo or no treatment (3 RCTs, n = 378; odds ratio (OR) 2.59, 95% CI 1.52 to 4.40; *p* < 0.001, *I*^2^ = 78%) with very low certainty evidence on GRADE assessment. Pooled analyses of data from various antioxidants suggest that antioxidants may improve live birth rates in women attending fertility clinics, whether or not they undergo MAR, compared with placebo or no treatment/standard treatment (13 RCTs, n = 1227; OR 1.81, 95% confidence interval (CI) 1.36 to 2.43; *p* < 0.001, *I*^2^ = 29%) [44]. However, the evidence is of very low certainty. A single RCT reported in Showell 2020 that NAC improved live birth rates in women with PCOS undergoing ovarian drilling compared to placebo [44]. The nutrients L-arginine, coQ10, melatonin, vitamin B complex, vitamin D, and vitamin E did not increase live birth rates. The evidence for CoQ10 and melatonin was very low certainty, but GRADE assessments for the other nutrients could not be conducted as each were reported in only one trial. There were no reviews on either folic acid or zinc that met the inclusion criteria.

### 3.5. Efficacy of Nutrient Supplementation for Clinical Pregnancy Rate

L-arginine, L-carnitine, coQ10, melatonin, myo-inositol, NAC, vitamin B complex, vitamin C, D and E, and multiple micronutrients were assessed for increased clinical pregnancy rate compared with placebo or no treatment. The following nutrient supplements may increase clinical pregnancy rates.

L-carnitine supplementation increased clinical pregnancy rates in women with clomiphene resistant PCOS when compared with placebo (2 RCTs, n = 450; OR 11.14, 95% CI 5.70 to 21.81; *p* < 0.001, *I*^2^ = 85%; very low certainty evidence) alongside clomiphene with or without metformin.CoQ10 supplementation increased clinical pregnancy rates in women undergoing MAR or with clomiphene resistant PCOS in comparison with placebo or no treatment (4 RCTs, n = 397; OR 2.49, 95% CI 1.50 to 4.13; *p* < 0.001, *I*^2^ = 47%; very low certainty evidence).Melatonin increased clinical pregnancy rates in women undergoing MAR when compared with placebo or no treatment (7 RCTs, n = 678; OR 1.66, 95% CI 1.12 to 2.47; *p* = 0.01, *I*^2^ = 0%; very low certainty evidence).Myo-inositol led to higher clinical pregnancy rates in women with PCOS compared with metformin (2 RCTs, n = 220; RR = 1.52, 95% CI: 1.05 to 2.18; *p* = 0.03, *I*^2^ = 3%; low certainty evidence).NAC supplementation in women with PCOS led to higher clinical pregnancy rates compared with no treatment (2 RCTs, n = 177; OR 2.15, 95% CI 1.01 to 4.60; *p* = 0.93, *I*^2^ = 0%; moderate certainty evidence) or with placebo (8 RCTs, n = 1601; OR 2.14, 95% CI 1.05 to 4.37, *p* < 0.01, *I*^2^ = 74%; very low certainty evidence).Vitamin D supplementation increased clinical pregnancy rate in women undergoing MAR when compared to placebo (9 RCTs, n = 1677; OR 1.49, 95% CI 1.05 to 2.11, *p* = 0.02, *I*^2^ = 54%; very low certainty evidence).

The nutrients L-arginine, vitamin B complex and vitamins C and E did not increase clinical pregnancy rates, with very low certainty evidence. There was only one RCT for each nutrient except for vitamin C, which had two RCTs. There were no reviews on either folic acid or zinc that met the inclusion criteria.

### 3.6. Efficacy of Nutrient Supplementation for Biochemical Pregnancy Rate

Vitamin D did not increase biochemical pregnancy rate in women undergoing MAR compared with placebo or no treatment [45]. However, the evidence is of very low certainty. No other reviews reported on biochemical pregnancy rate.

### 3.7. Safety of Nutrient Supplementation (Adverse Effects)

Data were limited for the three categories of adverse effects, miscarriage, multiple pregnancy, and ectopic pregnancy, in the included reviews. NAC, vitamin D, and pooled antioxidants had no effect on miscarriage rates or multiple pregnancy rates in women trying to conceive spontaneously or utilising MAR, with low to very low certainty evidence. Pooled antioxidants had no effect on ectopic pregnancy rates in women trying to conceive spontaneously or utilising MAR, with low certainty evidence. Adverse effects were not reported for the other nutrient supplements.

### 3.8. Certainty of Evidence

There was an absence of high-quality evidence according to GRADE assessments for the outcomes. Certainty in the overall estimates was low to very low for live birth, biochemical and clinical pregnancy rates, as well as for adverse effects including multiple pregnancies, miscarriage and ectopic pregnancies. The certainty was mostly downgraded due to serious risk of bias, very serious inconsistency and very serious imprecision. GRADE results for each comparison and outcome are outlined in Table 1.

## 4. Discussion

To our knowledge, this is the first umbrella review to critically assess top tier evidence on the efficacy and safety of nutrient supplements for female infertility. Overall, the evidence is very uncertain about the effect of multiple micronutrients and antioxidants on live birth rates, whereas L-arginine, coQ10, melatonin, vitamin B complex, vitamin D, and vitamin E did not increase live birth rates. Clinical pregnancy rates may be improved with L-carnitine, coQ10, melatonin, myo-inositol, NAC and vitamin D although the evidence is very uncertain, but not with L-arginine, vitamin B complex and vitamins C or E. Finally, biochemical pregnancy rate was not increased with vitamin D and was not assessed as an outcome in trials of other nutrients. While adverse effects were poorly reported, there is low to very low certainty that the use of NAC, vitamin D and antioxidants did not lead to more miscarriages, multiple or ectopic pregnancies. Across most of the assessed outcomes, the available evidence was of very low certainty due to the limited number and quality of included studies; hence, results should be interpreted with caution.

For live birth rates, we found very low certainty evidence of the benefit of multiple micronutrients. However, in the three RCTs examining these multiple micronutrients, the products and ingredients varied, including the dosages, making it difficult to draw valid conclusions or offer general recommendations. Similarly, there was very low certainty evidence that pooled antioxidants increased live birth rates. Antioxidants, biological and chemical compounds that reduce oxidative damage, are a diverse group of organic nutrients including vitamins, minerals and polyunsaturated fatty acids. These were administered either as single antioxidants or combined therapies making it difficult to compare interventions and provide general recommendations. A recent literature review indicated that multiple micronutrient supplementation in both healthy women and those with infertility, may offer benefits by helping restore micronutrients to recommended levels and reducing oxidative stress [29]. This highlights the need for further high-quality RCTs to determine the efficacy and safety of multiple micronutrients for live birth rates, both in women trying to conceive naturally and those using MAR.

L-carnitine, myoinositol and NAC demonstrated potential benefits in improving clinical pregnancy for women with PCOS but the evidence is very uncertain. These nutrients have distinct but essential functions in metabolism and reproductive health. L-carnitine plays an important role in metabolic and free fatty acid transport [48], while NAC is a source of sulfhydryl groups and a potent scavenger of reactive oxygen species [49,50]. Both mechanisms explain the potential benefits of these nutrients in enhancing cellular energy balance and reducing oxidative stress, thereby improving reproductive function. Inositol, derived from fruits and beans, is a second messenger for hormones such as follicle stimulating hormone (FSH), which has a key role in sexual development and reproduction [51]. Myo-inositol has been shown to reduce insulin resistance and androgens and improve ovarian function in women with PCOS, all of which would positively influence clinical pregnancy rate [52]. It should be noted that our findings are based on very few studies, with each nutrient examined in only two RCTs (n = 177–450). The evidence is therefore very uncertain and highlights a clear need for additional studies to determine whether, and to what extent, these nutrients can improve clinical pregnancy rates in women with PCOS in the context of female infertility.

In women undergoing MAR, CoQ10 may improve clinical pregnancy rate in our analysis of four RCTs (n = 397) with an odds ratio of 2.49 but the evidence is very uncertain. As a lipid-soluble structure, the primary role of CoQ10 is as an intermediate of the electron transport system in mitochondria, but also as an antioxidant to maintain redox homeostasis [53]. These properties have potential benefits for reversing ovarian dysfunction, promoting ovulation and oocyte maturation, and optimising embryonic development [53]. While data from preclinical and clinical studies indicate that CoQ10 supplementation is highly safe and well tolerated (at doses of 1200 mg/day) [54], further large-scale studies are needed to improve certainty in the estimates with respect to the efficacy of CoQ10 for achieving clinical pregnancy.

Finally, melatonin and vitamin D may improve clinical pregnancy rates in those undergoing MAR but the evidence is very uncertain. These nutrients had the largest number of studies with seven and nine RCTs of 678 to 1677 women for melatonin and vitamin D, respectively. Melatonin is a naturally occurring peptide hormone secreted by the pineal gland and multiple extra-pineal tissues (e.g., uterus, ovaries, and placenta). Its antioxidant and anti-inflammatory actions are thought to promote successful reproduction through slowing ovarian ageing [55]. Indeed, circulating melatonin levels gradually decline with age, along with a presumed loss of melatonin production in the mitochondria of the ovarian cells, coinciding with reduced oocyte quality as women age [55]. While evidence suggests a good safety profile, evidence certainty was very low due to very serious ROB, inconsistency, and imprecision. Thus, further research is needed to confirm the efficacy and safety of melatonin, especially with higher doses and longer treatment durations [56,57]. For vitamin D, which is derived primarily from exposure to ultraviolet radiation, its well-known actions relate to bone mineralisation and maintaining calcium and phosphorus levels [58]. There is evidence that vitamin D modulates male and female reproductive function [59], with vitamin D receptors expressed in numerous tissues of reproductive organs, such as ovaries and endometria [60]. HOXA10, a homeotic gene regulated by the interaction of sex steroid hormones and vitamin D [61], is an endometrial receptivity marker involved in embryo implantation [62]. Diminished HOXA10 expression can impair uterine receptivity and is implicated in decreased implantation associated with endometriosis, PCOS, and leiomyomas [62]. Our findings indicate improvements in clinical pregnancy rate with vitamin D, particularly in four studies that used calcitriol [45,63], the most active form of vitamin D, which functions similar to a steroid hormone [64] and regulates HOXA10 expression [61]. Based on the available evidence, vitamin D dosage in the range of 2000–4000 IU/d (50 to 100 μg/d) is safe for long-term use [65], however, the relationship between female reproduction and vitamin D in its different forms needs to be further explored.

Based on the available evidence, there are currently no specific nutrient supplements which can be confidently recommended for women aiming to conceive. Nevertheless, many of the studied interventions are of relatively low risk and may provide additional benefits to individuals with infertility such as addressing deficiencies and increasing antioxidant levels. While the reporting of adverse events was somewhat limited, the available evidence on CoQ10, melatonin, NAC, vitamin D, L-carnitine and inositol suggests that the supplements are safe and well tolerated. There is a need for improved reporting of adverse events from RCTs in order to inform clinical practice regarding long-term safety.

### Limitations

The main limitation of this umbrella review is that, although potential benefits were seen for some interventions in relation to improved female fertility outcomes, the overall certainty of evidence was very low to low. Our search was restricted to English language articles and grey literature (such as non-indexed reports or white papers) was not included, which limits generalisability. As we did not conduct a network meta-analysis, our findings should not be used for comparing efficacy rates between nutrients not compared in pair-wise meta-analyses.

Notwithstanding these limitations, this is the first umbrella review exploring the efficacy and safety of nutrient supplements for female infertility. Our review synthesises the most recent, hierarchical, and highest ranked evidence using a systematic approach that aims to limit overlap of primary studies. We used validated tools and processes, such as AMSTAR-2 and GRADE, to assess quality and confidence in effect estimates. We also incorporated assessments of adverse events with the aim of establishing a comprehensive profile of both the efficacy and safety of nutrient supplements in the context of fertility. In this process, we identified a clear need for more high quality RCTs to achieve higher confidence in the reported findings. In particular, clinically relevant outcomes such as live births and biochemical pregnancy were lacking, as were studies on adverse effects. Future RCTs should standardise dosages, types and administration durations of nutrient interventions to facilitate comparisons between studies. Where possible, studies should be conducted in a range of different populations to improve their external validity, allowing generalisability to a wider range of patients. Investigating the cost of nutrient supplements is also warranted. These aspects are critical to establishing definitive safety profiles and facilitating clinical decision-making for improved outcomes in women with infertility.

## 5. Conclusions

The available evidence is currently insufficient to recommend nutrient supplementation to improve female infertility due to the very low quality of evidence investigating the efficacy and safety of this therapy. However, there is currently no indication that these nutrients pose any risk of significant harm. There is a need for further high-quality RCTs to identify both the efficacy and safety of nutrient supplements for female fertility in women, trying to conceive naturally and in women using MAR.

## Data Availability

The authors confirm that the data supporting the findings of this study are available within the manuscript and its Appendix A.

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
