# Peer review of "The Role of Nutrient Supplements in Female Infertility: An Umbrella Review and Hierarchical Evidence Synthesis"

_nutrients, 2024, doi:10.3390/nu17010057_

Round 1

Reviewer 1 Report

Comments and Suggestions for Authors

This review is coined an ‘umbrella review’ since it aims at reviewing various reviews, in order to evaluate the role of nutrient supplements in primary outcomes (live birth rates and biochemical pregnancy successes) and secondary outcomes (miscarriage, ectopic pregnancy)

A spectacular observation is the drastic reduction in the number of studies on which the present paper is based, basically, as shown in figure 1, it started from 2378 non-duplicated papers to reach only 4.

1.       A major issue is to understand in more detail how the selection was made. In the first step, 2250 papers were excluded through reading their abstract. What justifies the choice? And in the 128 remaining, the reasons are exposed but it is generally not clear to the reader what they really mean except some that are readily understandable such as ‘non-English language’. What is ‘wrong outcome’? what is ‘wrong study design’? Does it mean a basic flaw in the exposure? In the analysis? If so how come that the initial paper was published? What is ‘Wrong patient population’? is it because of an ethnic background? Is it because the end point phenotype was not correctly described?

2.       A second major issue is the overall result is that very low certainty was obtained for any of the antioxidant or other nutrient was used.

3.       This leads to a major issue. If the initial screening off the study would have been less stringent, and thus would include much more different studies, then would it be possible to get results for some nutrients with a better certainty? So, the question is ‘is it possible to decrease the stringency that currently maintained 0.16% of the studies to reach may be 1-5% of the studies and see whether better certainties could be obtained. Currently, the major outcome of the paper is only a negative conclusion that is that supplementation is not known to have any effect. If changing the threshold has no effect then, this could be the definitive conclusion of the present study. But at this step, it seems that a less stringent re-analysis would be warranted.

Minor point:

1.       Why in the secondary outcomes was IUGR not included?

Author Response

Nutrients 3340184

The role of nutrient supplements in female infertility: an umbrella review and hierarchical synthesis

Thank you for the opportunity to submit a revised manuscript and note that the feedback has been useful in strengthening this manuscript.

Reviewer 1 Comments:
This review is coined an ‘umbrella review’ since it aims at reviewing various reviews, in order to evaluate the role of nutrient supplements in primary outcomes (live birth rates and biochemical pregnancy successes) and secondary outcomes (miscarriage, ectopic pregnancy)

A spectacular observation is the drastic reduction in the number of studies on which the present paper is based, basically, as shown in figure 1, it started from 2378 non-duplicated papers to reach only 4.

  1. A major issue is to understand in more detail how the selection was made. In the first step, 2250 papers were excluded through reading their abstract. What justifies the choice? And in the 128 remaining, the reasons are exposed but it is generally not clear to the reader what they really mean except some that are readily understandable such as ‘non-English language’. What is ‘wrong outcome’? what is ‘wrong study design’? Does it mean a basic flaw in the exposure? In the analysis? If so how come that the initial paper was published? What is ‘Wrong patient population’? is it because of an ethnic background? Is it because the end point phenotype was not correctly described?

On page 3 lines 124-152 we describe the eligibility criteria that determine the studies to be included that will answer our research question “Which nutrients are efficacious and safe for improving fertility outcomes in females with infertility trying to conceive spontaneously or utilising MAR?” These criteria outline the participants, intervention, control and outcomes that are relevant to our research question. Excluding reviews is not a reflection of any judgement on our part about the rigour or the value of the review. It is only a decision that the review does not assist us in answering our research question. For example “wrong outcome” means that the review did not report on live birth rate, clinical or biochemical pregnancies, miscarriages, ectopic or multiple pregnancies, “wrong study design” means it was not a systematic review/meta-analysis/umbrella review/network meta-analysis, “wrong patient population” means it was not in females with infertility trying to conceive spontaneously or using medically assisted reproduction.

  1. A second major issue is the overall result is that very low certainty was obtained for any of the antioxidant or other nutrient was used.

We have no control over whether the overall result is of very low or high certainty. By definition we are including reviews that analysed already published randomised controlled trials and the features of these trials that may increase or decrease certainty of the evidence (number of trials, sample sizes, risk of bias etc – as per the GRADE approach) is beyond our control. This lack of certainty has been highlighted transparently in the limitations section of our discussion on page 15 lines 409-433, emphasising the need for further rigorous primary research to improve evidence certainty. We believe this is an important finding on its own, and very low certainty evidence should not guide whether a study is published or not, as study design and conduct should be determined a priori and studies should be published regardless of a positive or negative result, and regardless of degree of certainty of the findings.

  1. This leads to a major issue. If the initial screening off the study would have been less stringent, and thus would include much more different studies, then would it be possible to get results for some nutrients with a better certainty? So, the question is ‘is it possible to decrease the stringency that currently maintained 0.16% of the studies to reach may be 1-5% of the studies and see whether better certainties could be obtained. Currently, the major outcome of the paper is only a negative conclusion that is that supplementation is not known to have any effect. If changing the threshold has no effect then, this could be the definitive conclusion of the present study. But at this step, it seems that a less stringent re-analysis would be warranted. 

Criteria were decided on a priori as per best practice in review methodology. It is not possible to decrease stringency to try and include more reviews. An umbrella review is an overview of systematic reviews, with the unit of inclusion being systematic reviews and not RCTs. Including more reviews will in fact not change the certainty of the evidence but either duplicate the same evidence or add outdated and poorer quality studies to decrease (not increase) certainty.

We aim to include the most recent and highest level of evidence for each individual nutrient, to avoid duplication of findings (see lines 154-167 re hierarchical evidence synthesis method). We already have criteria that mean we are searching for all nutrient supplements that reported on our primary and secondary outcomes of live births, clinical and biochemical pregnancies, miscarriage, ectopic and multiple pregnancies which are considered clinically important outcomes. These will assist us in answering our question of which nutrient supplements are efficacious for improving live birth and clinical pregnancy rate in females with infertility. It is impossible for us to create research findings from RCTs, this is not the purpose of our umbrella review; we can only report on what has been published and what meets our PICOS criteria. We note also that the Cochrane review (Showell et al) is a large review (63 RCTs) covering 17 supplements and therefore while we have included 4 reviews, this covers 11 nutrients.

       4. Minor point:

           Why in the secondary outcomes was IUGR not included?

In the methods we defined the secondary outcomes in lines 148-149. Secondary outcomes were: “Any adverse effects reported by the review including miscarriage and ectopic or multiple pregnancy.” None of the included reviews investigated intrauterine growth restriction (IUGR) as an outcome, therefore it was not included in our overview.

Reviewer 2 Report

Comments and Suggestions for Authors

nutrients-3340184-peer-review-v1

Overview

The study seems well-conducted and the manuscript is well-written about the worldwide concern.

The reviewer has some questions.

Critical comments

I am interested in Table 1, where folate is not listed as a single low/column (included in MMN). Does this mean that there is no meta-analysis on folate alone that meet the authors’ inclusion criteria? If so, the description like “there is no meta-analysis on folate alone that meet the authors’ inclusion criteria” in Discussion will help potential readers understand the situation.

Similarly, zinc is not listed as a single low/column (included in MMN and Vitacap). Does this mean that there is no meta-analysis on zinc alone that meet the authors’ inclusion criteria? If so, the description like “there is no meta-analysis on zinc alone that meet the authors’ inclusion criteria” in Discussion will help potential readers.

The same thing goes for astaxanthin. If so, the description like “there is no meta-analysis on zinc alone that meet the authors’ inclusion criteria” in Discussion will help potential readers.

Were there any studies on flavonols (kaempferol, quercetin, kaempferitrin, and isorhamnetin) that authors’ inclusion criteria?

I also have an interest in randomized controlled trial on Trapa bispinosa Roxb extract improved oocyte developmental potential and endometrial receptivity, increased live birth rate in older patients undergoing artificial reproductive technology (Jinno M, Nagai R, Takeuchi M, Watanabe A, Teruya K, Sugawa H, Hatakeyama N, Jinno Y. Reprod Biol Endocrinol. 2021 Sep 27;19(1):149.). Does this study meet the authors’ inclusion criteria?

Author Response

Nutrients 3340184

The role of nutrient supplements in female infertility: an umbrella review and hierarchical synthesis

Thank you for the opportunity to submit a revised manuscript and note that the feedback has been useful in strengthening this manuscript.

Reviewer 2 Comments:

  1. I am interested in Table 1, where folate is not listed as a single low/column (included in MMN). Does this mean that there is no meta-analysis on folate alone that meet the authors’ inclusion criteria? If so, the description like “there is no meta-analysis on folate alone that meet the authors’ inclusion criteria” in Discussion will help potential readers understand the situation.

This is correct. There were no systematic review/meta-analyses/umbrella reviews (SR/MA/UR) on folic acid/folate that met our inclusion criteria. There was only one review on folate in the list of studies that were excluded at full-text review, and it referred to women’s folate levels rather than supplementation. A sentence has been added to the Results Section 3.4 Efficacy of nutrient supplementation for live birth rate (lines 274-275) and Section 3.5 Efficacy of nutrient supplementation for clinical pregnancy rate (lines 303-304). It reads:

There were no reviews on either folic acid or zinc that met the inclusion criteria.”

In the introduction (page 2 lines 96-99), it was noted that current guidelines “have made recommendations for improving preconception health, however, supplementation is limited to folic acid 35, intended to prevent neural tube disorders rather than to assist conception”.

Also one review stated that based on clinical advice folic acid < 1mg was considered ‘standard treatment’. A sentence has been added to the Results Section 3.2.3 Control. It reads:

Folic acid at doses less than 1mg were considered ‘standard treatment’ in Showell et al. 2020. 44

  1. Similarly, zinc is not listed as a single low/column (included in MMN and Vitacap). Does this mean that there is no meta-analysis on zinc alone that meet the authors’ inclusion criteria? If so, the description like “there is no meta-analysis on zinc alone that meet the authors’ inclusion criteria” in Discussion will help potential readers.

This is also correct. There were no SR/MA/UR on zinc that met our inclusion criteria. Zinc was included as a keyword in our search strategy in Supplementary material Table S1.

A sentence has been added to the Results Section 3.4 Efficacy of nutrient supplementation for live birth rate (lines 274-275) and Section 3.5 Efficacy of nutrient supplementation for clinical pregnancy rate (lines 303-304). It reads:

There were no reviews on either folic acid or zinc that met the inclusion criteria.”

  1. The same thing goes for astaxanthin. If so, the description like “there is no meta-analysis on zinc alone that meet the authors’ inclusion criteria” in Discussion will help potential readers.

There was no SR/MA/UR for astaxanthin that met our inclusion criteria. It was not, however, part of our search strategy (described in 2.1 page 3 lines 114-117) which was developed by the authors. “A preliminary scoping search was conducted to identify reviews on the topic, noting text words and index terms in relevant reviews.” Since there was no evidence of the use of astaxanthin for conception, it is not appropriate to mention this particular nutrient in the discussion as suggested.

  1. Were there any studies on flavonols (kaempferol, quercetin, kaempferitrin, and isorhamnetin) that authors’ inclusion criteria?

There were no SR/MA/URs for any of these flavonols that met our inclusion criteria.

  1. I also have an interest in randomized controlled trial on Trapa bispinosa Roxb extract improved oocyte developmental potential and endometrial receptivity, increased live birth rate in older patients undergoing artificial reproductive technology (Jinno M, Nagai R, Takeuchi M, Watanabe A, Teruya K, Sugawa H, Hatakeyama N, Jinno Y. Reprod Biol Endocrinol. 2021 Sep 27;19(1):149.). Does this study meet the authors’ inclusion criteria?

This umbrella review addressed the use of nutrient supplements for fertility outcomes and our inclusion criteria specified that herbal medicines were excluded (Page 3 lines 142-143) and also RCTs are excluded as the included studies are only systematic reviews and higher.  This RCT assessing the herbal medicine Trapa bispinosa Roxb would have been excluded.

Round 2

Reviewer 2 Report

Comments and Suggestions for Authors

The manuscript improved adequately.